# Outcome Analysis in Elective Electrical Cardioversion of Atrial Fibrillation Patients: Development and Validation of a Machine Learning Prognostic Model

**DOI:** 10.3390/jcm11092636

**Published:** 2022-05-07

**Authors:** Jean C. Nuñez-Garcia, Antonio Sánchez-Puente, Jesús Sampedro-Gómez, Victor Vicente-Palacios, Manuel Jiménez-Navarro, Armando Oterino-Manzanas, Javier Jiménez-Candil, P. Ignacio Dorado-Diaz, Pedro L. Sánchez

**Affiliations:** 1Department of Cardiology, Hospital Universitario de Salamanca—IBSAL, 37007 Salamanca, Spain; nunezjean@hotmail.com (J.C.N.-G.); jesusmanuelsg@gmail.com (J.S.-G.); victor.vicente.palacios@philips.com (V.V.-P.); aoterino1@hotmail.com (A.O.-M.); jimenezcandil@secardiologia.es (J.J.-C.); pedro.dorado@gmail.com (P.I.D.-D.); 2CIBERCV (Centro de Investigacion Biomedica en Red Enfermedades Cardiovasculares), Instituto de Salud Carlos III, C/Monforte de Lemos 3-5, Pabellón 11, Planta 0, 28029 Madrid, Spain; 3Philips Healthcare, 28050 Madrid, Spain; 4Department of Cardiology, Hospital Virgen de la Victoria—IBIMA, 29010 Malaga, Spain; mjimeneznavarro@gmail.com; 5Facultad de Medicina, Universidad de Málaga, 29071 Malaga, Spain; 6Departamento de Medicina, Universidad de Salamanca, 37007 Salamanca, Spain

**Keywords:** machine-learning, electrical cardioversion, atrial fibrillation, rhythm control, pharmacologic cardioversion

## Abstract

Background: The integrated approach to electrical cardioversion (EC) in atrial fibrillation (AF) is complex; candidates can resolve spontaneously while waiting for EC, and post-cardioversion recurrence is high. Thus, it is especially interesting to avoid the programming of EC in patients who would restore sinus rhythm (SR) spontaneously or present early recurrence. We have analyzed the whole elective EC of the AF process using machine-learning (ML) in order to enable a more realistic and detailed simulation of the patient flow for decision making purposes. Methods: The dataset consisted of electronic health records (EHRs) from 429 consecutive AF patients referred for EC. For analysis of the patient outcome, we considered five pathways according to restoring and maintaining SR: (i) spontaneous SR restoration, (ii) pharmacologic-cardioversion, (iii) direct-current cardioversion, (iv) 6-month AF recurrence, and (v) 6-month rhythm control. We applied ML classifiers for predicting outcomes at each pathway and compared them with the CHA2DS2-VASc and HATCH scores. Results: With the exception of pathway (iii), all ML models achieved improvements in comparison with CHA2DS2-VASc or HATCH scores (*p* < 0.01). Compared to the most competitive score, the area under the ROC curve (AUC-ROC) was: 0.80 vs. 0.66 for predicting (i); 0.71 vs. 0.55 for (ii); 0.64 vs. 0.52 for (iv); and 0.66 vs. 0.51 for (v). For a threshold considered optimal, the empirical net reclassification index was: +7.8%, +47.2%, +28.2%, and +34.3% in favor of our ML models for predicting outcomes for pathways (i), (ii), (iv), and (v), respectively. As an example tool of generalizability of ML models, we deployed our algorithms in an open-source calculator, where the model would personalize predictions. Conclusions: An ML model improves the accuracy of restoring and maintaining SR predictions over current discriminators. The proposed approach enables a detailed simulation of the patient flow through personalized predictions.

## 1. Introduction

Restoring and maintaining sinus rhythm (SR) is an integral part of the atrial fibrillation (AF) process. Electrical cardioversion (EC) quickly and effectively converts AF to SR and can be performed safely for patients with AF of ≥48 h or unknown duration when anticoagulation with vitamin-K antagonists, a factor Xa inhibitor, or a direct thrombin inhibitor is used for at least 3 weeks before and at least 4 weeks after EC [1,2,3].

Although EC restores SR in around 80% of patients, the rate of recurrence is high—around 60% in the coming months [4,5,6,7,8,9,10]—even under antiarrhythmic drugs [11,12,13,14,15,16]. Furthermore, it has been described that up to 60% of patients with recent-onset AF and candidates for rhythm control resolve spontaneously even while waiting for scheduled EC [17]. Thus, the integrated approach to elective EC is complex, and it is especially interesting to identify potential predictors of recurrence post-cardioversion, in order to avoid unnecessary drugs or procedures that could involve risks and costs in addition to avoiding the programming of EC or the use of drugs in patients who would restore the SR spontaneously. For this purpose, traditional clinical models have been previously proposed [18,19,20,21,22,23,24], although their use and utility in clinical practice is unclear due to the complexity of AF management.

Interest in machine-learning (ML) in electrophysiology is increasing in order to enhance automatic clinical workflows and increase efficiency [25]. Although ML is starting to be widely applied in arrhythmia [26], examples regarding the whole process workflow for a clinician to make better decisions are scarce [27]. In this study, we used ML to move the EC of AF process management a notch ahead. AF patients go through different pathways: from the diagnosis of the AF and prescription of anticoagulation and antiarrhythmic drugs to the post-cardioversion medical follow-up. We analyzed the whole elective EC of the AF process using ML algorithms, in order to enable a more realistic and detailed simulation of the patient flow for decision making purposes. This study followed the TRIPOD guidelines for reporting the development and validation of prognostic models [28], see Appendix A.

## 2. Materials and Methods

Figure 1 summarizes the phases we followed to build our ML models: preparation of the model, model training, and model evaluation. The models were developed in Python and the implementation of the classification algorithms was performed using the open code libraries scikit-learn and xgboost [29].

### 2.1. Preparation of the Model

#### 2.1.1. Task Definition and Clinical Pathways of Patients

The aim of our study was to automatically enhance the process of scheduled EC in AF by incorporating ML in all pathways of the process to predict success. In pursuing a rhythm-control strategy, patients scheduled for planned EC followed a process that is summarized in Figure 2, where the different outcomes at each pathway have been highlighted. Importantly, management options in hemodynamically stable patients with AF >48 h in our hospital do not follow the strategy of treatment guided by transesophageal echocardiography findings [1,2].

Given the outcomes at each pathway, we aimed to build an ML model for each circumstance: (i) spontaneous SR restoration, predicting the conversion to SR in the pre-scheduled EC period for non-antiarrhythmics-treated patients; (ii) pharmacologic cardioversion, predicting the conversion to SR in the pre-scheduled EC period for antiarrhythmics-treated patients; (iii) direct-current cardioversion, predicting the efficacy of direct-current shock application; (iv) AF recurrence, predicting the AF recurrence at the 6-month follow-up for those patients who underwent SR restoration spontaneously, by pharmacologic or direct-current cardioversion; and (v) rhythm control, predicting the overall 6-month follow-up maintenance in SR from the moment EC was scheduled.

#### 2.1.2. Study Population

From April 2014 to January 2019, a registry of 429 consecutive patients scheduled for planned EC in the tertiary referral university hospital of Salamanca were included in the analysis. EC in our center relies on the application of direct-current biphasic waveform shock, with a fixed energy of 150 J with a progressive energy level of 200 J, via two anteroposterior (parasternal and left infrascapular) electrodes. Patients undergo the procedure in our Cardiology Day Hospital by trained personnel, usually under propofol sedation, and remain under observation for at least 3 h before discharge, where an ECG is performed to check the heart rhythm [30]. For all the patients, a visit to the outpatient clinic was scheduled at 6-months, where a second ECG was also performed. Implantable loop recorders or Holter ECGs were not used either before or after the scheduled cardioversion.

#### 2.1.3. Data Collection and Preparation

The ML models were trained and validated with the use of the patient charts stored in electronic health records (EHRs). Input data (features) consisted of patient demographics, cardiovascular risk factors, cardiovascular history, comorbidities, clinical and biochemical variables, atrial fibrillation classification, echocardiographic findings, medical treatment, and direct-current shock variables. As for the corresponding outcomes, we labeled the presence of SR in 4 of the analyzed pathways (spontaneous restoration of SR, pharmacologic cardioversion, direct-current cardioversion, and 6-month rhythm control) and the presence of AF for the 6-month AF recurrence. All EHRs were reviewed by a single investigator who classified the type of AF according to the current guidelines in paroxysmal AF, persistent AF, and long-standing persistent AF [1,2].

We preprocessed our EHR raw data as a set of features to be usable by ML classifiers and a set of labels to classify the different outcomes for each of the patients. For this purpose, multicategory variables were one-hot encoded in binary variables. Missing data were imputed using the average of the rest of the dataset for continuous variables and the median for categorical variables. Weight and height were imputed according to gender specific averages, and if only weight was missing, BMI was imputed first, then weight was obtained from BMI and height. The value of tricuspid regurgitant jet velocity was imputed using the average of likewise severity of tricuspid regurgitation patients in the dataset. The dataset was divided then into a training dataset consisting of 316 patients that attended before 1 January 2018 and a testing dataset consisting of 113 patients that attended afterwards.

### 2.2. Training the Model

#### 2.2.1. Machine Learning Classifiers

The goal of the training phase was to produce a working ML model that accepted data from any new patient (formatted in the same way as our processed dataset) and classified it. We applied and compared the performance of the following state-of-the-art ML classifiers: logistic regression with a regularization term, random forest, extremely randomized trees, and boosted trees [26].

#### 2.2.2. Hyperparameter Tuning

Model hyperparameters are the properties that govern the behavior of the classification algorithm, i.e., the number of branches in a boosted trees algorithm. Tuning these parameters may improve the performance of the ML models and was consequently conducted in our pipeline.

To determine the best performing hyperparameters without using the testing dataset, a stratified cross-validation scheme was used. We performed a 10-fold cross-validation methodology to randomly split the training dataset into 10 equally sized parts (folds), with equal distribution of positive and negative cases. Nine folds were used to train the algorithms with different combinations of hyperparameters, and the remaining one was used as a test dataset for evaluating the models. We used these predictions to choose the best hyperparameters for each classification algorithm (Table 1).

### 2.3. Evaluating the Model

#### 2.3.1. Evaluation Scheme

The models were evaluated on the test dataset. Additionally, internal validation was performed using the training dataset only. This internal validation consisted of a stratified 10-fold cross-validation with 10 repetitions. Since the training of the model also contained a hyperparameter tuning step with its own cross-validation scheme, this resulted in nested cross-validations [31]. The information from this internal validation was used to transform the models into hard classifiers to be used in clinical practice by choosing a probability cutoff threshold that translated continuous probability predictions into distinct clinical decisions.

In both the internal and external validation, the receiver-operating-characteristic (ROC) and the Precision-Recall (PR) curve analysis were used to assess the predictive capacity of the ML models at each clinical pathway [32]. The classification performance of the model at a particular cutoff threshold was evaluated according to its sensitivity (recall), specificity, positive predictive value (precision), and negative predictive value. Confidence intervals were calculated for both the external validation results [33] and the internal validation results. The latter ones were calculated using a t-statistic based on the fold results, corrected for the correlation between fold samples [34,35].

#### 2.3.2. Comparison with Standard Successful Cardioversion Risk Scores

We further compared the performance of the developed ML algorithms to existing predictive multivariate logistic regression models: CHA2DS2-VASc [23] and HATCH [36,37] scores. For this comparison, we evaluated the existing scores directly on our dataset, essentially performing an external validation of the prediction rules. In order not to give the ML models an unfair advantage, we further refitted the scores with beta coefficients in our study population for the different pathways’ outcomes. In addition, we estimated the Net Reclassification Index (NRI) of the ML models with respect to the existing scores, calculated at the optimum cutoff threshold for the score [38]. This index was the difference of the sum of sensitivity and specificity between two classifiers.

#### 2.3.3. Feature Analysis

The differences in data variables between event and non-event patient groups in each AF pathway were compared using χ^2^ or Fisher tests for categorical variables and Student’s *t*-test or ANOVA for continuous variables.

We further computed feature importance for the models by measuring how the area under the ROC curve (AUC ROC) decreased when a feature was not available through the method known as permutation importance or mean decrease accuracy (MDA) [39]. The method consisted of replacing each feature in the test dataset with random noise-feature column and measuring the performance for the ML model. The weight of the feature with positive impact in the predictive model was scaled to 1. This method was chosen because it is classification algorithm-agnostic and offers an intuitive idea of what happens when some part of the data of a given subject is missing and is substituted by a random value distributed according to the rest of the population.

#### 2.3.4. Open-Source Software

The developed code used to train and evaluate the models can be consulted as open-source at https://github.com/IA-Cardiologia-husa/Cardioversion, (accessed on 7 April 2022) [40]. We deployed our ML classifiers in an online open-source calculator that can be run on any Google Drive account, as an example tool for prospective external validation of ML models from a small imbalanced sample size. The calculator chained all ML models to provide personalized outcome predictions.

## 3. Results

### 3.1. Characteristics and Flow of the Study Population

The characteristics of the study population are shown in Table 2. The EHRA classification of atrial fibrillation symptoms was not widely described in the EHRs, and it was not provided. Figure 2 presents the movement of patients through the elective EC of the AF process. The presence of SR on the ECG recorded at the end of the scheduled EC visit occurred in 374 (87.2%) of the 429 patients included in the study: in 52 (20.6%) of 252 non-antiarrhythmics-treated patients, conversion to SR occurred spontaneously in the pre-scheduled EC period; in 35 (19.8%) of 177 antiarrhythmics-treated patients, pharmacologic cardioversion occurred in the pre-scheduled EC period; and of the 342 patients still in AF at the scheduled EC visit, 287 (83.9%) converted to SR after direct-current shock application. Among the 374 patients in SR after the scheduled-EC visit, a recurrence of AF occurred in 145 (38.8%) patients on the ECG recorded at the 6-month visit. Thus, final successful rhythm control at 6 months was achieved in 229 (53.4%) of the 429 patients initially included in the study.

### 3.2. Comparison of Prediction Models for Each Pathway

The prediction accuracy of the different models under consideration evaluated at each clinical pathway is shown in Table 3 and Table 4 for the cross-validation with training data and the evaluation with testing data, respectively. We used both the CHA2DS2-VASc and HATCH risk scores as baseline models for performance evaluation. With the exception of the direct-current cardioversion pathway, all the standard ML models achieved statistically significant improvements compared to the baseline CHA2DS2-VASc or HATCH scores (*p* < 0.01).

The best overall ML classifier algorithm in the internal validation was extremely randomized trees with an AUC ROC of 0.81 for spontaneous SR restoration, 0.68 for pharmacological cardioversion, 0.47 for direct-current cardioversion, 0.67 for 6-month AF recurrence, and 0.69 for overall 6-month rhythm control, and it was chosen as the classifier algorithm to be used with the test set and the online open-source calculator.

In order to better assess the clinical significance of these results, we compared the classification performance in the 113 patients test set of the ML model with the CHA2DS2-VASc and HATCH risk scores, operating at an optimal threshold that was selected based on the ROC and PR curves.

For the spontaneous restoration of SR or pharmacologic cardioversion, the ML model classified 35 patients as likely to return to SR before the scheduled direct-current shock application (Figure 3A). Of those, 16 returned to SR before the direct-current shock application (46% precision); meanwhile, of the 78 remaining patients, 68 stayed in AF (87% negative predictive value). For the direct-current cardioversion (Figure 3B), the ML model classified 83 patients as likely to be in SR after the electric shock, and 4 with likely to remain in AF; 73/83 of the likely to be in the SR group returned to SR (88% precision), and so did 2/4 of the likely to remain in the AF group (50% negative predictive value). For the recurrence of AF (Figure 3C), out of the 101 patients that returned to SR, the ML model grouped them as 30 likely to have a recurrence within 6 months and 71 not likely to have a recurrence. From the likely to have a recurrence group, 16/30 did (53% precision); meanwhile, from the not likely to have a recurrence group, 47/71 stayed in SR (66% negative predictive value). Finally, for the overall success of rhythm control at 6 months (Figure 3D), the ML categorized the 113 patients into 69 patients likely to be successful and 44 patients not likely to be successful. In the likely to be successful group, 45/69 were in SR (65% precision); meanwhile in the not likely to be successful group, 28/44 were in AF (64% negative predictive value).

Compared to the most competitive existing score (Table 5), the ML model classified correctly three more patients in the positive class and six less in the negative class for predicting spontaneous SR restoration, for an NRI of +5.9% in favor of the ML model; it classified correctly 2 less patients in the positive class and 22 more patients in the negative class for predicting pharmacologic cardioversion, for an NRI of +38.8% in favor of the ML model; it classified correctly 12 more patients in the positive class and 2 less in the negative class for predicting direct-current cardioversion, for an NRI of −0.6% favoring the HATCH score; it predicted correctly two patients more in the positive class and six patients more in the negative class for predicting 6-month AF recurrence, for an NRI of +14.8% in favor of the ML model; and finally, it predicted four more patients in the positive class and eight more in the negative class for predicting the overall 6-month rhythm control, for an NRI of +22.1% in favor of the ML model.

### 3.3. Feature Importance

Table 6 shows the five most important variables, along with their importance scores, ranked according to their contribution to the predictions of the ML model at each pathway. Variables related to paroxysmal AF classification and left atrial dilatation appeared to be more important for the predictions than traditional cardiovascular risk factors and age included in the CHA2DS2-VASc or HATCH scores. Paroxysmal AF was on the list of top predictors of three pathways: spontaneous SR restoration, pharmacologic cardioversion, and 6-month rhythm control, while left atrial dilatation was among the most important risk factors of four pathways: spontaneous SR restoration, direct-current cardioversion, AF recurrence, and 6-month rhythm control.

### 3.4. Machine-Learning Models Deployment in a Calculator

We deployed our machine-learning algorithms in a calculator where you can input the information of the 18 features that were found as main predictors for each clinical pathway and see the individual prediction for the concrete outcome. These features are weight, height, time of AF onset, LA volume, mitral regurgitation, LVEF, NYHA functional class, tobacco smoking history, previous direct-current shock application attempt, previous transient ischemic attack or stroke, history of heart failure, history of anticoagulation, pulmonary disease including sleep apnea, impaired physical mobility, beta blockers, ACE inhibitors/Angiotensin receptor blockers, and type of anticoagulation. For the prediction of AF recurrence, additional features, such as the type of cardioversion (spontaneous, pharmacologic, or direct current), antiarrhythmic prescription, and creatinine clearance, were required. The calculator is available at https://colab.research.google.com/drive/1TbHf9waHNQYHQJhu5M9iqnpO5AESGDO5, accessed on 7 April 2022.

## 4. Discussion

To our knowledge, this is the first ML analysis of the whole elective EC of the AF process. In a consecutive and well-characterized cohort, we were able to concatenate different ML algorithms to establish predictions at each different clinical pathway observed throughout the EC process. Our ML prediction models were superior to the classical existing scores, CHA2DS2-VASc and HATCH. Taking into account the difficulty of using ML algorithms in clinical practice, we further integrated them into a simple open-source calculator where predictions are easy to calculate and understand.

Other investigators have used the ML methodology to predict particular pathways of the EC of the AF process, such as Oto et al. studying the successful cardioversion workflow for patients who perform pharmacologic cardioversion after 48 h of flecainide treatment [42] or Sterling et al. predicting successful cardioversion using ECG variables [21]. However, we would like to highlight that the analysis presented here is more integral than the previous ones, that we have considered ML models with nonlinear interactions between the variables, and that we have been more thorough with the feature selection phase to avoid pitfalls in our evaluation phase.

In our study, the performance of the resulting ML models ranged from a very good classification for the spontaneous restoration of SR and the pharmacological cardioversion models; a reasonable classification for the AF recurrence and successful cardioversion model; to a not statistically significantly better than random guessing classification for the direct-current cardioversion model. Differences in performance between models with different ML classification algorithms in each of the workflows were not statistically significant, and we reported the best algorithm result among them. The results from the development were consistent with the ones from the validation.

The classical existing predictive multivariate logistic regression scores, CHA2DS2-VASc and HATCH, are currently considered the cornerstone for the management of AF. Although the CHA2DS2-VASc score is the basis for the management of anticoagulation therapy [1,2], some studies suggest that it has predictive value for AF recurrence after cardioversion. In a pooled meta-analysis collecting data of 2889 patients, the CHA2DS2-VASc score was an independent predictor of early recurrence of AF after pharmacologic or EC [23]. In addition, the HATCH score, initially described to predict progression from paroxysmal to persistent AF [36], has also been shown to be useful in predicting the short-term success of EC [37]. The herein developed ML models have been consistently better than the CHA2DS2-VASc and HATCH scores at predicting the different outcomes in our cohort of patients. We must acknowledge that the existing scores are facing an external validation, meanwhile ML models are evaluated internally, and the possible selection bias or other existing biases in our dataset will play in favor of the ML models. However, the results of both existing scores were poor and unlikely to be useful in clinical practice; meanwhile, the results of the ML models were optimal in the independent dataset, corresponding to 113 patients, where they were validated. An external validation should be performed to confirm that these results that could be improved by using additional datasets for the continuous development of the models. In particular, a bigger dataset or the inclusion of different variables, such as ECG variables, might help in discovering additional interactions between features with predictive properties. This is the main reason why open-source of the developed algorithms can be accessed through this publication, with the aim of improving their predictions with the addition of new patient cohorts and new variables.

The clinical application of the ML prediction models is relevant. Using the developed open-source calculator, we could make individual predictions for each AF patient for whom EC is a therapy option and optimize the procedure (i.e., adding pre-cardioversion antiarrhythmic drugs) in cases with a low likelihood of having a successful cardioversion, or we would prioritize the waiting list for those patients for whom the EC is estimated to be successful over time. The use of the developed open-source calculator is simple and can facilitate the implementation of other elective EC models where nursing plays a predominant role [43].

Finally, this study has several limitations. The results of the ML models in the five AF workflows have been uneven. In particular, we have not been able to predict success of the application of direct current cardioversion. The results of the recurrence of AF, and subsequently the long-term success of the cardioversion process, were also moderate. We must acknowledge that we are working with a single-hospital dataset and that it would have been desirable to have a different population to validate the ML models. Nevertheless, we consider it enough to showcase the possibilities of applying ML techniques in a clinical workflow and want to emphasize the effort made to offer a rigorous evaluation, performing nested cross-validation steps for selecting features and hyperparameters to report an accurate measure of the performance. A greater number of patients would have allowed for the development of more precise models and to study in more detail the relationship between variables and outcomes. However, the sample size and number of events were enough to perform a proper evaluation of the ML and is reflected in that we were able to ascertain statistically significant differences in performance between ML models and risk scores (Figure 4). We encourage researchers with larger databases to use the provided code as a basis to build more refined models.

## Figures and Tables

**Figure 1 jcm-11-02636-f001:**
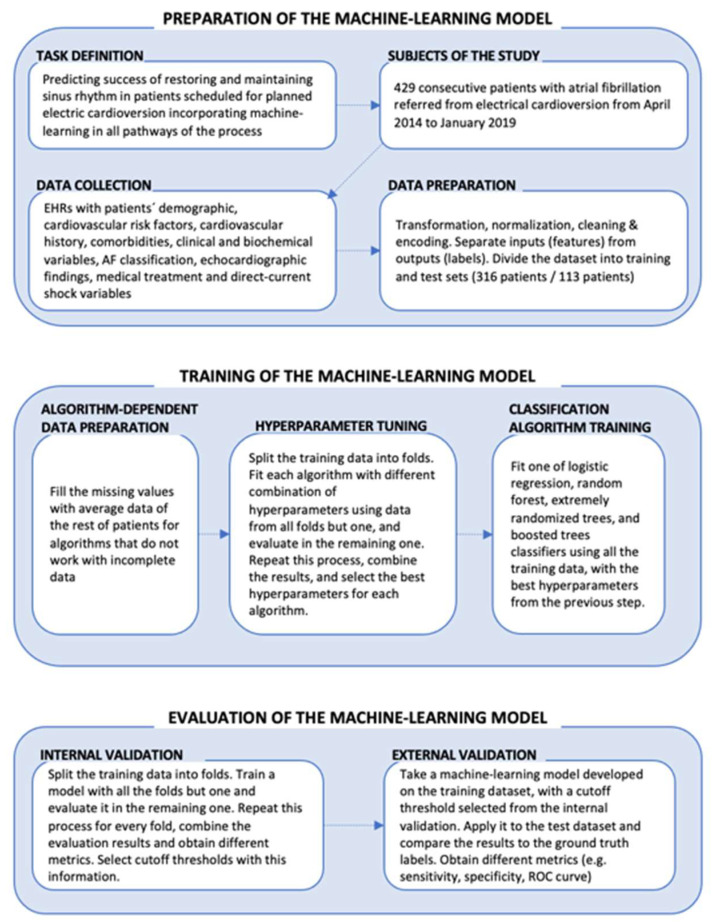
Overview of the phases followed to build and evaluate the machine-learning models.

**Figure 2 jcm-11-02636-f002:**
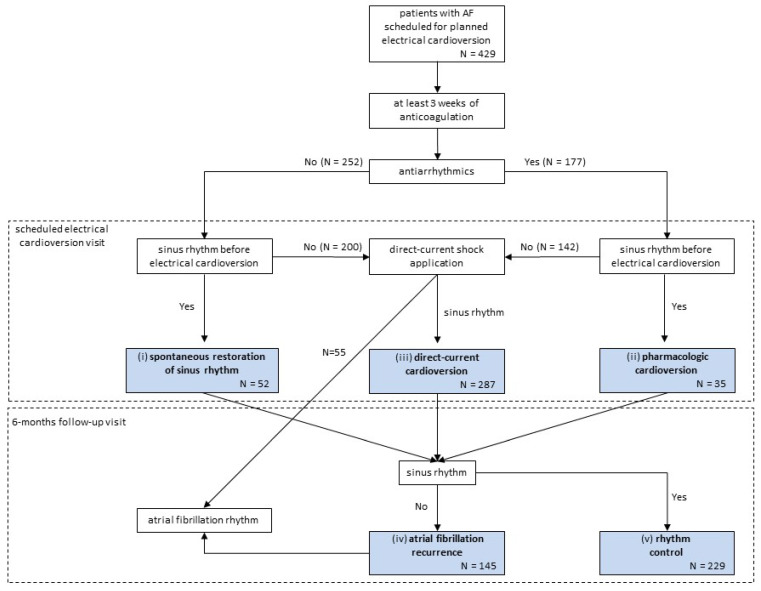
Patients scheduled for planned electrical cardioversion flow diagram where the different outcomes at each pathway are highlighted. The different machine-learning models were then built for each of these 5 different circumstances: (i) spontaneous sinus rhythm restoration (conversion to sinus rhythm in the pre-scheduled electrical cardioversion period for non-antiarrhythmics-treated patients); (ii) pharmacologic cardioversion (conversion to sinus rhythm in the pre-scheduled electrical cardioversion period for antiarrhythmics-treated patients); (iii) direct-current cardioversion (conversion to sinus rhythm after direct-current shock application); (iv) atrial fibrillation recurrence (atrial fibrillation recurrence at 6-month follow-up for those patients who underwent sinus rhythm restoration spontaneously, by pharmacologic or direct-current cardioversion); and (v) rhythm control (maintenance in sinus rhythm at 6-month follow-up).

**Figure 3 jcm-11-02636-f003:**
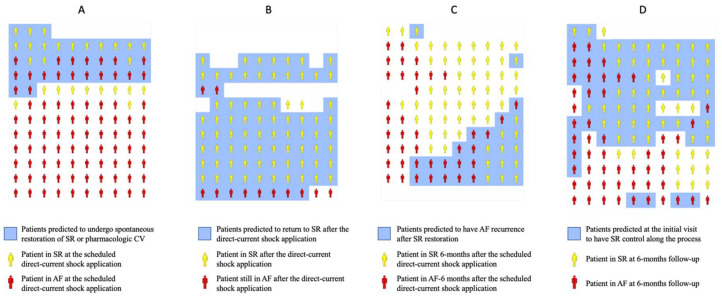
Illustration of envisioned clinical utilization of the machine-learning predictions along the elective electrical cardioversion (EC) process. For the predictions, it was used an independent dataset (from that used for the generation of the machine-learning models) of 113 patients. Patients in sinus rhythm (SR) are represented in yellow and patients in atrial fibrillation (AF) in red. Patients predicted by the machine-learning model to undergo or be in SR are included in a blue background. Panel (**A**) represents predictions (blue background) to undergo spontaneous restoration of SR or pharmacological cardioversion (CV) and ground truth findings for each patient (yellow or red). Panel (**B**) represents predictions (blue background) of efficacy of direct-current shock application and ground truth findings for each dataset patient (yellow or red). Panel (**C**) represents predictions (blue background) of AF recurrence at 6 months after SR restoration and ground truth findings for each patient (yellow or red). Panel (**D**) represents predictions (blue background) of SR control at 6 months and ground truth findings for each patient (yellow or red).

**Figure 4 jcm-11-02636-f004:**
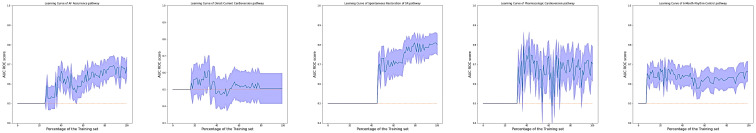
The learning curve of the best model for each of the pathways, including hyperparameter tuning. The results are shown as the area under the ROC curve with its confidence interval as measured in the external validation set. Notice that the results of the models are not displayed until a certain percentage of the training set is used. This is because the hyperparameter tuning step performs a cross validation that requires a minimum of events to produce results.

**Table 1 jcm-11-02636-t001:** Hyperparameters tested during the tuning step. This table contains the different combinations of feature selection strategies and hyperparameters tested for each of the classification algorithms during training.

Algorithm	Feature Selection	Hyperparameters
Boosted Trees	No selection Univariate selection RF feature importance	Number of trees: 25, 100, or 1000 Depth of the trees: 3, 5, or 7 Learning rate: 0.1 or 0.05 L1 regularization term weights: 0 or 1 L2 regularization term weight. 1
Random Forest	No selection Univariate selection Feature importance (random forest)	Number of trees: 100 or 1000 Number of features considered at each branch split: 1 or auto (square root of total features) Split criterion: Gini impurity or information gain. Max depth of trees: 1, 2, 5, or unbounded
Extremely Randomized Trees	No selection Univariate selection RF feature importance	Number of trees: 100 or 1000 Number of features considered at each branch split: 1 or auto (square root of total features) Split criterion: Gini impurity or information gain. Max depth of trees: 1, 2, 5, or unbounded
Logistic Regression	No selection Univariate selection RF feature importance	Regularization term: L1 or L2

**Table 2 jcm-11-02636-t002:** Baseline characteristics of the study cohort. List of continuous and categorical data input of the patients used for ML model development. Continuous variables are expressed as mean ± standard deviation and categorical as *n* (%). Reference ranges for LVEF were considered normal greater than 50%, mild dysfunction from 49 to 40%, moderate dysfunction from 39 to 30%, and severe dysfunction less than 30%. Paroxysmal AF was defined as AF with episodes recurring with variable frequency; persistent AF was defined as continuous AF that is sustained >7 days; long-standing persistent AF was defined as continuous AF >12 months in duration. Reference ranges for LA volume index were considered normal <35 mL/m^2^, mildly dilated from 35 to 41 mL/m^2^, moderately dilated from 42 to 48 mL/m^2^, and severely dilated >48 mL/m^2^ [41].

	Development Dataset N = 316		Validation Dataset N = 113	
	Missing Values	Mean	Missing Values	Mean
**Demographics**				
Age, years	2	62.1 ± 11.9	4	63.0 ± 12.2
Gender, male	-	240 (75.9%)	-	83 (73.5%)
Weight, kg	30	84.0 ± 17.1	19	85.2 ± 20.1
Height, cm	39	170.0 ± 8.9	28	170.6 ± 9.9
Body mass index, kg/m^2^	41	28.9 ± 5.0	28	29.0 ± 6.2
**Cardiovascular risk factors**				
Hypertension	-	172 (54.4%)	-	62 (54.9%)
Dyslipidemia	-	128 (40.5%)	-	47 (41.6%)
Active smoking	-	46 (14.6%)	-	12 (10.6%)
Smoking history	-	133 (42.1%)	-	45 (39.8%)
Diabetes mellitus	-	59 (18.7%)	-	23 (20.4%)
**Cardiovascular history**				
Heart failure	-	100 (31.6%)	-	24 (21.2%)
Coronary artery disease	-	52 (16.5%)	-	11 (9.7%)
Previous direct-current shock application attempt	-	22 (7.0%)	-	15 (13.3%)
Previous transient ischemic attack or stroke	-	19 (6.0%)	-	6 (5.3%)
History of oral anticoagulation treatment	-	158 (50.0%)	-	33 (29.2%)
Peripheral vascular disease	-	16 (5.1%)	-	9 (8.0%)
Rheumatic heart disease	-	7 (2.2%)	-	1 (0.9%)
**Other comorbidities**				
Chronic obstructive pulmonary disease	-	66 (20.9%)	-	15 (13.3%)
Prior cancer	-	28 (8.9%)	-	10 (8.8%)
Prior bleeding	-	11 (3.5%)	-	3 (2.7%)
Venous thromboembolism	-	10 (3.2%)	-	1 (0.9%)
Impaired physical mobility	-	8 (2.5%)	-	6 (5.3%)
**Clinical and biochemical variables**				
NYHA functional class >I	-	106 (33.5%)	-	39 (34.5%)
NYHA functional class >II	-	35 (11.1%)	-	10 (8.8%)
NYHA functional class >III	-	9 (2.8%)	-	0.0 ± 0.0
CHAD2DS2-VASc score	-	2.2 ± 1.7	-	2.1 ± 1.6
HATCH score	-	1.6 ± 1.5	-	1.4 ± 1.2
HASBLED score	-	2.3 ± 1.1	-	2.1 ± 0.9
Anemia	-	35 (11.1%)	-	14 (12.4%)
Creatinine, mg/dL	-	1.0 ± 0.4	1	1.0 ± 0.3
Glomerular filtration rate, mL/min/1.73 m^2^	-	75.6 ± 17.1	1	77.0 ± 17.0
**Atrial fibrillation classification**				
Paroxysmal	-	61 (19.3%)	-	24 (21.2%)
Persistent	-	250 (79.1%)	-	89 (78.8%)
Long-standing persistent	-	5 (1.6%)	-	0 (0%)
**Echocardiographic findings**				
LV mass index, g/m^2^	42	102.3 ± 32.4	53	97.7 ± 28.4
LVEF < 50%	-	73 (23.1%)	-	26 (23.0%)
LVEF < 40%	-	42 (13.3%)	-	13 (11.5%)
LVEF < 30%	-	21 (6.6%)	-	6 (5.3%)
Tricuspid regurgitant jet velocity, cm/sec	149	257.7 ± 47.1	67	247.0 ± 50.9
At least moderate probability pulmonary hypertension	-	43 (13.6%)	-	9 (8.0%)
High probability pulmonary hypertension	-	10 (3.2%)	-	2 (1.8%)
LA volume index, mL/m^2^	37	43.6 ± 17.4	44	44.9 ± 16.7
LA volume index ≥ 35 mL/m^2^	-	182 (57.6%)	-	51 (45.1%)
LA volume index ≥ 42 mL/m^2^	-	138 (43.7%)	-	34 (30.1%)
LA volume index > 48 mL/m^2^	-	98 (31.0%)	-	27 (23.9%)
Significant valvular heart disease	-	68 (21.5%)	-	21 (18.6%)
Mitral stenosis	-	2 (0.6%)	-	1 (0.9%)
Mitral regurgitation	-	39 (12.3%)	-	15 (13.3%)
Aortic stenosis	-	2 (0.6%)	-	3 (2.7%)
Aortic regurgitation	-	9 (2.8%)	-	4 (3.5%)
Tricuspid regurgitation	-	23 (7.3%)	-	5 (4.4%)
Mechanical prosthetic valve	-	10 (3.2%)	-	1 (0.9%)
Biological prosthetic valve	-	9 (2.8%)	-	5 (4.4%)
**Oral anticoagulation**				
Time under anticoagulation, days	-	30.9 ± 23.2	-	27.4 ± 17.2
K-vitamin antagonist	-	102 (32.3%)	1	14 (12.5%)
Direct oral anticoagulants	-	214 (67.7%)	1	98 (87.5%)
Dabigatran	-	23 (7.3%)	-	11 (9.7%)
Rivaroxaban	-	93 (29.4%)	-	17 (15.0%)
Apixaban	-	79 (25.0%)	-	49 (43.4%)
Edoxaban	-	19 (6.0%)	-	20 (17.7%)
Low-weight-molecular heparin	-	0 (0%)	-	2 (1.8%)
**Antiarrhythmic drugs**				
Antiarrhythmics before scheduled EC	-	132 (41.8%)	-	45 (39.8%)
Amiodarone before scheduled EC	-	98 (31.0%)	-	34 (30.1%)
Flecainide before scheduled EC	-	30 (9.5%)	-	11 (9.7%)
Dronedarone before scheduled EC	-	4 (1.3%)	-	0 (0%)
Antiarrhythmics after scheduled EC	-	198 (62.7%)	-	65 (57.5%)
Amiodarone after scheduled EC	-	147 (46.5%)	-	38 (33.6%)
Flecainide after scheduled EC	-	45 (14.2%)	-	27 (23.9%)
Dronedarone after scheduled EC	-	6 (1.9%)	-	0 (0%)
**Concomitant medications**				
Nonsteroidal anti-inflammatory drug	-	5 (1.6%)	-	0 (0%)
Aspirin	-	39 (12.3%)	-	7 (6.2%)
Dual antiplatelet therapy	-	4 (1.3%)	-	1 (0.9%)
Beta-blocker	-	243 (76.9%)	-	88 (77.9%)
ACE inhibitors/angiotensin II receptor blocker	-	155 (49.1%)	-	40 (35.4%)
Sacubitril-Valsartan	-	2 (0.6%)	-	1 (0.9%)
Calcium antagonist	-	50 (15.8%)	-	11 (9.7%)
Aldosterone receptor antagonist	1	31 (9.8%)	-	10 (8.8%)
Digoxin	-	20 (6.3%)	-	1 (0.9%)
**Direct-current procedure**				
Number of shocks	76	1.4 ± 0.7	30	1.3 ± 0.6
Applied maximal energy, J	105	176.2 ± 102.6	42	165.6 ± 36.5

ACE = angiotensin converting enzyme; EC = electric cardioversion; LA = left atrial; LVEF = left ventricle ejection fraction.

**Table 3 jcm-11-02636-t003:** Performance of all prediction models at each clinical pathway in the cross-validation of the training data, measured in terms of the area under the ROC curve (AUC ROC) and area under the precision-recall curve (AUC PR). Both the CHA2DS2-VASc and HATCH risk scores were used as baseline models for the performance evaluation of each machine-learning developed model.

Pathway	Predictions	Model	AUC-ROC	AUC-ROC Change		AUC-PR	AUC-PR Change	
Spontaneous SR restoration	1840	CHA2DS2-VASc	0.62 (0.50–0.73)	Baseline model	−7%	0.33 (0.23–0.44)	Baseline model	−2%
		HATCH	0.69 (0.58–0.80)	+7%	Baseline model	0.35 (0.25–0.45)	+2%	Baseline model
		Regularized logistic regression	0.81 (0.71–0.92)	+19%	+12%	0.68 (0.53–0.82)	+35%	+33%
		Random forest	0.82 (0.72–0.92)	+20%	+13%	0.67 (0.53–0.81)	+34%	+32%
		Extremely randomized trees	0.81 (0.71–0.92)	+19%	+12%	0.68 (0.54–0.83)	+35%	+33%
		Boosted trees	0.80 (0.70–0.91)	+18%	+11%	0.68 (0.53–0.82)	+35%	+33%
Pharmacologic cardioversion	1320	CHA2DS2–VASc	0.53 (0.39–0.67)	Baseline model	−2%	0.29 (0.20–0.37)	Baseline model	+2%
		HATCH	0.55 (0.43–0.67)	+2%	Baseline model	0.27 (0.21–0.33)	−2%	Baseline model
		Regularized logistic regression	0.74 (0.60–0.87)	+21%	+19%	0.64 (0.47–0.80)	+35%	+37%
		Random forest	0.67 (0.49–0.85)	+14%	+12%	0.60 (0.42–0.77)	+31%	+33%
		Extremely randomized trees	0.68 (0.51–0.84)	+15%	+13%	0.58 (0.41–0.75)	+29%	+31%
		Boosted trees	0.68 (0.53–0.84)	+15%	+13%	0.61 (0.45–0.78)	+32%	+34%
Direct-current cardioversion	2550	CHA2DS2-VASc	0.52 (0.42–0.62)	Baseline model	–6%	0.85 (0.81–0.89)	Baseline model	−1%
		HATCH	0.58 (0.47–0.68)	+6%	Baseline model	0.86 (0.82–0.90)	+1%	Baseline model
		Regularized logistic regression	0.51 (0.40–0.62)	−1%	−7%	0.85 (0.80–0.89)	0%	−1%
		Random forest	0.48 (0.38–0.59)	−4%	−10%	0.85 (0.80–0.89)	0%	−1%
		Extremely randomized trees	0.47 (0.35–0.58)	−5%	−11%	0.84 (0.79–0.88)	−1%	−2%
		Boosted trees	0.46 (0.38–0.55)	−6%	−12%	0.84 (0.80–0.87)	−1%	−2%
6-month AF recurrence	2730	CHA2DS2-VASc	0.54 (0.47–0.61)	Baseline model	−4%	0.40 (0.35–0.46)	Baseline model	+2%
		HATCH	0.58 (0.50–0.65)	+4%	Baseline model	0.38 (0.33–0.43)	−2%	Baseline model
		Regularized logistic regression	0.63 (0.55–0.71)	+9%	+5%	0.55 (0.47–0.63)	+15%	+17%
		Random forest	0.67 (0.59–0.75)	+13%	+9%	0.61 (0.52–0.70)	+21%	+23%
		Extremely randomized trees	0.68 (0.61–0.75)	+14%	+10%	0.61 (0.52–0.70)	+21%	+23%
		Boosted trees	0.63 (0.55–0.71)	+9%	+5%	0.57 (0.48–0.65)	+17%	+19%
6-month rhythm control	3160	CHA2DS2-VASc	0.55 (0.48–0.62)	Baseline model	−4%	0.58 (0.52–0.63)	Baseline model	−2%
		HATCH	0.59 (0.52–0.69)	+4%	Baseline model	0.60 (0.54–0.66)	+2%	Baseline model
		Regularized logistic regression	0.63 (0.57–0.70)	+8%	+4%	0.69 (0.63–0.74)	+11%	+9%
		Random forest	0.68 (0.62–0.74)	+13%	+9%	0.71 (0.65–0.77)	+13%	+11%
		Extremely randomized trees	0.69 (0.62–0.75)	+14%	+10%	0.72 (0.65–0.78)	+14%	+12%
		Boosted trees	0.57 (0.51–0.64)	+2%	−2%	0.63 (0.58–0.68)	+5%	+3%

Number of predictions = Number of samples × 10 repetition.

**Table 4 jcm-11-02636-t004:** Performance of all prediction models at each clinical pathway in the evaluation with testing data. Both the CHA2DS2-VASc and HATCH risk scores were used as baseline models for the performance evaluation of each machine-learning developed model.

Pathway	Predictions	Model	AUC-ROC	AUC-ROC Change		AUC-PR	AUC-PR Change	
Spontaneous SR restoration	68	CHA2DS2-VASc	0.57 (0.59–0.65)	Baseline model	−9%	0.31 (0.24–0.39)	Baseline model	−7%
		HATCH	0.66 (0.59–0.73)	+9%	Baseline model	0.38 (0.30–0.47)	+7%	Baseline model
		Regularized logistic regression	0.80 (0.75–0.86)	+23%	+14%	0.52 (0.44–0.60)	+21%	+14%
		Random forest	0.72 (0.66–0.79)	+15%	+6%	0.48 (0.39–0.56)	+17%	+10%
		Extremely randomized trees	0.79 (0.73–0.84)	+22%	+13%	0.57 (0.49–0.64)	+26%	+19%
		Boosted trees	0.77 (0.71–0.83)	+20%	+11%	0.56 (0.48–0.64)	+25%	+18%
Pharmacologic cardioversion	45	CHA2DS2-VASc	0.45 (0.34–0.56)	Baseline model	−10%	0.18 (0.09–0.27)	Baseline model	−5%
		HATCH	0.55 (0.45–0.66)	+10%	Baseline model	0.23 (0.13–0.33)	+5%	Baseline model
		Regularized logistic regression	0.62 (0.52–0.72)	+17%	+7%	0.43 (0.32–0.54)	+25%	+20%
		Random forest	0.66 (0.57–0.76)	+21%	+11%	0.40 (0.29–0.51)	+22%	+17%
		Extremely randomized trees	0.71 (0.63–0.80)	+26%	+16%	0.42 (0.31–0.53)	+24%	+19%
		Boosted trees	0.57 (0.46–0.67)	+12%	+2%	0.30 (0.19–0.40)	+12%	+7%
Direct-current cardioversion	87	CHA2DS2-VASc	0.57 (0.48–0.66)	Baseline model	+2%	0.88 (0.81–0.94)	Baseline model	+1%
		HATCH	0.55 (0.46–0.65)	–2%	Baseline model	0.87 (0.81–0.94)	–1%	Baseline model
		Regularized logistic regression	0.53 (0.44–0.62)	–4%	–2%	0.87 (0.81–0.94)	–1%	0%
		Random forest	0.41 (0.32–0.49)	–16%	–14%	0.85 (0.77–0.92)	–3%	–2%
		Extremely randomized trees	0.48 (0.39–0.57)	–9%	–7%	0.88 (0.81–0.94)	0%	+1%
		Boosted trees	0.58 (0.48–0.67)	+1%	+3%	0.91 (0.86–0.97)	+3%	+4%
6-month AF recurrence	101	CHA2DS2-VASc	0.52 (0.46–0.58)	Baseline model	+1%	0.41 (0.35–0.47)	Baseline model	+1%
		HATCH	0.51 (0.45–0.56)	–1%	Baseline model	0.40 (0.34–0.46)	–1%	Baseline model
		Regularized logistic regression	0.64 (0.59–0.70)	+12%	+13%	0.49 (0.43–0.55)	+8%	+9%
		Random forest	0.61 (0.55–0.67)	+9%	+10%	0.50 (0.44–0.56)	+9%	+10%
		Extremely randomized trees	0.62 (0.56–0.68)	+10%	+11%	0.53 (0.47–0.59)	+12%	+13%
		Boosted trees	0.57 (0.51–0.63)	+5%	+6%	0.48 (0.42–0.54)	+7%	+8%
6-month rhythm control	113	CHA2DS2-VASc	0.50 (0.45–0.56)	Baseline model	–1%	0.54 (0.48–0.59)	Baseline model	0%
		HATCH	0.51 (0.46–0.56)	+1%	Baseline model	0.54 (0.49–0.60)	0%	Baseline model
		Regularized logistic regression	0.66 (0.61–0.71)	+16%	+15%	0.68 (0.63–0.73)	+14%	+14%
		Random forest	0.60 (0.54–0.65)	+10%	+9%	0.62 (0.56–0.67)	+8%	+8%
		Extremely randomized trees	0.60 (0.55–0.65)	+10%	+9%	0.61 (0.56–0.67)	+7%	+7%
		Boosted trees	0.58 (0.53–0.63)	+8%	+7%	0.63 (0.58–0.68)	+9%	+9%

Number of predictions = Number of samples × 10 repetition.

**Table 5 jcm-11-02636-t005:** Classification analysis. The classification performance of the CHA2DS2-VASc and HATCH risk scores and the best performance machine-learning model were calculated for each electric cardioversion pathway. The net increase performance (number of patients and percentage) and net reclassification index were provided when utilizing the developed machine-learning model. The most competitive existing risk score, either CHA2DS2-VASc or HATCH, was used as the baseline model for the performance evaluation of the machine-learning developed model at each pathway.

Pathway/Model	TP	FP	TN	FN	R	S	P	NPV	Net Reclassification Index
**Spontaneous SR restoration**									
Extremely randomized trees	12	16	35	5	70.6%	68.6%	42.9%	87.5%	+5.9%
CHA2DS2-VASc ≤ 1	9	20	31	8	52.9%	60.8%	31%	79.5%	−19.6%
HATCH ≤ 0	9	10	41	8	52.9%	80.4%	47.4%	83.7%	Baseline model
**Pharmacologic cardioversion**									
Extremely randomized trees	4	3	33	5	44.4%	91.7%	57.1%	86.8%	+38.8%
CHA2DS2-VASc ≤ 2	6	25	11	3	66.7%	30.6%	19.4%	78.6%	Baseline model
HATCH ≤ 2	8	36	0	1	88.9%	0%	18.2%	0%	−8.4%
**Direct-current cardioversion**									
Extremely randomized trees	73	10	2	2	97.3%	16.7%	88%	50%	−0.6%
CHA2DS2-VASc ≤ 0	10	3	9	65	13.3%	75%	76.9%	12.1%	−26.3%
HATCH ≤ 1	61	8	4	14	81.3%	33.3%	88.4%	22.2%	Baseline model
**6-month AF recurrence**									
Extremely randomized trees	16	14	47	24	40%	77%	53.3%	66.2%	+14.8%
CHA2DS2-VASc >2	14	20	41	26	35%	67.2%	41.2%	61.2%	Baseline model
HATCH >1	15	23	38	25	37.5%	62.3%	39.5%	60.3%	−2.4%
**6-month rhythm control**									
Extremely randomized trees	45	24	28	16	73.8%	53.8%	65.2%	63.6%	+22.1%
CHA2DS2-VASc ≤ 2	41	32	20	20	67.2%	38.5%	56.2%	50%	Baseline model
HATCH ≤ 1	38	31	21	23	62.3%	40.4%	55.1%	47.7%	−2.8%

**Table 6 jcm-11-02636-t006:** Feature importance. Variable ranking by their contribution to the predictions of the extremely randomized tree model at each pathway. The score represents the relative importance of that variable for the machine-learning model. The weight of the features is scaled from 0 to 1; thus, variables close to 1 show a higher impact on the predictive model.

Pathway	Variable	Score
**Spontaneous SR restoration**	Paroxysmal atrial fibrillation	1
	History of oral anticoagulation treatment	0.316
	LA volume index ≥ 42 mL/m^2^	0.257
	ACE inhibitors/Angiotensin II receptor blockers	0.150
	LVEF < 50%	0.065
**Pharmacologic cardioversion**	Paroxysmal atrial fibrillation	1
	Heart failure	0.111
	Dyslipidemia	0.085
	Glomerular filtration rate	0.066
	Peripheral vascular disease	0.064
**Direct-current cardioversion**	Chronic obstructive pulmonary disease	1
	Long-standing persistent AF	0.693
	Heart Failure	0.411
	Beta blockers	0.297
	LA volume index ≥ 35 mL/m^2^	0.277
**6-month AF recurrence**	Spontaneous SR restoration	1
	History of oral anticoagulation treatment	0.857
	Hypertension	0.849
	ACE inhibitors/angiotensin II receptor blockers	0.827
	NYHA functional class >II	0.818
**6-month rhythm control**	LA volume index ≥ 35 mL/m^2^	1
	Paroxysmal atrial fibrillation	0.577
	History of oral anticoagulation treatment	0.468
	LA volume index ≥ 48 mL/m^2^	0.446
	Smoking history	0.423

LA = left atrial; LVEF = left ventricular ejection fraction; SR = sinus rhythm.

## Data Availability

Data supporting reported results and the developed code used to train and evaluate the models can be consulted as open-source at can be found, at https://github.com/IA-Cardiologia-husa/Cardioversion, accessed on 7 April 2022. The developed open-source calculator is available at https://colab.research.google.com/drive/1TbHf9waHNQYHQJhu5M9iqnpO5AESGDO5, accessed on 7 April 2022.

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
