# Peer review of "Outcome Analysis in Elective Electrical Cardioversion of Atrial Fibrillation Patients: Development and Validation of a Machine Learning Prognostic Model"

_jcm, 2022, doi:10.3390/jcm11092636_

Round 1

Reviewer 1 Report

The authors have analyzed the whole elective EC of AF process using machine-learning (ML) in order to enable more a realistic and detailed simulation of the patient flow for decision making purposes. The dataset consisted of electronic health records (EHRs) from 429 consecutive AF patients referred for EC. For analysis of the patient outcome, the authors considered 5-pathways according to restoring and maintaining SR: (i) spontaneous SR restoration, (ii) pharmacologic-cardioversion, (iii) direct-current cardioversion, (iv) 6-month AF recurrence and, (v) 6-month rhythm control. The authors applied ML classifiers for predicting outcomes at each pathway and compared them with the CHA2DS2-VASc and HATCH scores.
With the exception for pathway (iii), all ML models achieved improvements in comparison with CHA2DS2-VASc or HATCH scores (p <0.01). Compared to the most competitive score, the AUC-ROC was: 0.80 vs. 0.66 for predicting (i); 0.71 vs. 0.55 for (ii); 0.64 vs. 0.52 for (iv) and; 0.66 vs. 0.51 for (v). For a threshold considered optimal, the empirical net reclassification index was: 35 + 7.8%, + 47.2%, + 28.2% and + 34.3% in favor of our ML models for predicting outcomes for pathways (i), (ii), (iv ) and (v), respectively.

The article is well written and brings new knowledge to atrial fibrillation. Nevertheless, in the introduction it is worth emphasizing that in patients with atrial fibrillation it is worth considering additional statin therapy (Ntaios, G et al; Wańkowicz, P et al).

Author Response

RC: The authors have analyzed the whole elective EC of AF process using machine-learning (ML) in order to enable more a realistic and detailed simulation of the patient flow for decision making purposes. The dataset consisted of electronic health records (EHRs) from 429 consecutive AF patients referred for EC. For analysis of the patient outcome, the authors considered 5-pathways according to restoring and maintaining SR: (i) spontaneous SR restoration, (ii) pharmacologic-cardioversion, (iii) direct-current cardioversion, (iv) 6-month AF recurrence and, (v) 6-month rhythm control. The authors applied ML classifiers for predicting outcomes at each pathway and compared them with the CHA2DS2-VASc and HATCH scores.

With the exception for pathway (iii), all ML models achieved improvements in comparison with CHA2DS2-VASc or HATCH scores (p <0.01). Compared to the most competitive score, the AUC-ROC was: 0.80 vs. 0.66 for predicting (i); 0.71 vs. 0.55 for (ii); 0.64 vs. 0.52 for (iv) and; 0.66 vs. 0.51 for (v). For a threshold considered optimal, the empirical net reclassification index was: 35 + 7.8%, + 47.2%, + 28.2% and + 34.3% in favor of our ML models for predicting outcomes for pathways (i), (ii), (iv ) and (v), respectively.

The article is well written and brings new knowledge to atrial fibrillation. Nevertheless, in the introduction it is worth emphasizing that in patients with atrial fibrillation it is worth considering additional statin therapy (Ntaios, G et al; Wańkowicz, P et al).

AA: Dear reviewer, we have reviewed the suggested articles. Both are related to the additional beneficial prognosis of a statin in the primary or secondary prevention of atrial fibrillation stroke, and not related to any strategy to restore and maintain sinus rhythm in patients with atrial fibrillation. This is why we do not know how to fit both articles in our manuscript introduction. However, any suggestion from you is welcome to further integrate them.     

Reviewer 2 Report

Dear Sir/Madam,

I had the opportunity to act as a reviewer on the recent submission by Nuñez-Garcia et al. to the Journal of Clinical Medicine.

The authors present an interesting original article investigating the potential role of a machine learning model in predicting restoring and maintaining sinus rhythm in patients with atrial fibrillation. They have included 429 consecutive patients with atrial fibrillation referred for electrical cardioversion. The manuscript is well written.

However, some major issues need to be addressed:

  1. The sample size of 429 in order to test and develop the algorithm seems rather small. I strongly recommend at least doubling the cohort and then training the machine learning algorithm.
  2. There are no baseline data showing the EHRA classification of atrial fibrillation symptoms. In fact, the information regarding atrial fibrillation symptoms is completely missing. We recommend adding this to the manuscript.
  3. In the Materials and Method section, lines 106-107, the authors state: “For all the patients, a visit to the outpatient clinic 106 was scheduled at 6-months, where a second ECG was also performed.” This seems to introduce bias, as there is no information regarding to cardioversions or episodes of atrial fibrillation, which were treated in other hospitals. Furthermore, how was the atrial fibrillation burden before cardioverting assessed – ideally with means of implantable loop recorder or Holter ECG? Please clarify.
  4. Based on which criteria was the use of antiaarhythmic medication after the cardioversion employed?
  5. What does exactly “significant valvular disease” mean: did you include only mild and moderate valvular disease? Any symptomatic severe valvular heart disease should be treated before attempting cardioversion. Furthermore, I recommend excluding patients with valvular atrial fibrillation (i.e., moderate to severe mitral stenosis and mechanical prosthetic valve in mitral position).
  6. In the Discussion section, lines 384-388, the authors state: “Using the developed open-source calculator, we could make individual predictions for each AF patient in whom EC is a therapy option and avoid the procedure in cases with a low likelihood of having a successful cardioversion and opt for a rate control strategy, which has been significantly associated with increase risk of adverse events[42], or to avoid antiarrhythmic drugs in patients who have high likelihood of spontaneous restoration of SR”. I find this very problematic, since this should be always in clinical practice refined. For example, a highly symptomatic patient presenting with atrial fibrillation in whom the sinus rhythm cannot be restored, should undergo a trial of antiarrhythmic therapy and then a new cardioversion attempt. In case of recurrence under antiarrhythmics catheter ablation of atrial fibrillation must be discussed. Furthermore, avoiding antiarrhythmic drugs in patients who have a high likelihood of spontaneous restoration of SR is not the best strategy, because of the nature of atrial fibrillation to become permanent. As such, strategies such as risk factor management, antiarrhythmic therapy and ablation need to be considered.

Best regards,

Alexandru Bejinariu

Author Response

RC: I had the opportunity to act as a reviewer on the recent submission by Nuñez-Garcia et al. to the Journal of Clinical Medicine.

The authors present an interesting original article investigating the potential role of a machine learning model in predicting restoring and maintaining sinus rhythm in patients with atrial fibrillation. They have included 429 consecutive patients with atrial fibrillation referred for electrical cardioversion. The manuscript is well written.

However, some major issues need to be addressed:

    The sample size of 429 in order to test and develop the algorithm seems rather small. I strongly recommend at least doubling the cohort and then training the machine learning algorithm.

AA: We thank the reviewer for the detailed revision of the manuscript and the insightful comments and suggestions. We acknowledge that the sample size is small, and this is indeed a limitation of the study. A greater number of patients would have allowed to develop more precise models and to study in more the detail the relationship between variables and outcomes. However, the sample size and number of events was enough to perform a proper evaluation of the machine learning models and is reflected in that we were able to ascertain statistically significant differences in performance between our developed models and the risk scores. In the revised version, we have acknowledged the limitation in sample size and encouraging researchers with larger databases to use the provided code as a basis to build more refined models.

RC: There are no baseline data showing the EHRA classification of atrial fibrillation symptoms. In fact, the information regarding atrial fibrillation symptoms is completely missing. We recommend adding this to the manuscript.

AA: Dear reviewer, we appreciate your recommendation and we recognize the importance and relevance of the EHRA classification for our study. However, the EHRA classification was not widely described in medical records and due to the retrospective nature of data collection, we cannot include it in our study.

RC: In the Materials and Method section, lines 106-107, the authors state: “For all the patients, a visit to the outpatient clinic 106 was scheduled at 6-months, where a second ECG was also performed.” This seems to introduce bias, as there is no information regarding to cardioversions or episodes of atrial fibrillation, which were treated in other hospitals. Furthermore, how was the atrial fibrillation burden before cardioverting assessed – ideally with means of implantable loop recorder or Holter ECG? Please clarify.

AA: The University Hospital of Salamanca is the only hospital attending patients in the Province of Salamanca. In our EHRs records there were no information regarding other episodes of atrial fibrillation or cardioversion out of our institution. Implantable loop recorders or holter ECG were not used either before or after the scheduled cardioversion, as it is not clinical practice in our Hospital. This information has been added in the new version of the manuscript.

RC: Based on which criteria was the use of antiaarhythmic medication after the cardioversion employed?

AA: The use of antiarrhythmic medication before or after cardioversion was performed at the discretion of the treating cardiologist.

RC: What does exactly “significant valvular disease” mean: did you include only mild and moderate valvular disease? Any symptomatic severe valvular heart disease should be treated before attempting cardioversion. Furthermore, I recommend excluding patients with valvular atrial fibrillation (i.e., moderate to severe mitral stenosis and mechanical prosthetic valve in mitral position).

AA: This is an important point raised by the reviewer. The appearance of atrial fibrillation in a patient with a severe valvular heart disease is stated in most guidelines as a class IIA indication for valve repair/replacement. In our study, significant valve disease means any valvular disease, at least moderate for regurgitation, mild for stenosis, or the presence of a prosthetic valve. Table 2 shows the number of patients with these significant valvular diseases and their types in our study. The cardioversion was indicated because the valvular heart disease was not severe enough to be treated before attempting cardioversion. Furthermore, we did not want to exclude patients with prosthetic valves as we found of interest to study through machine learning whether they are related with atrial fibrillation recurrence. In contract of expected, significant valvular disease was not a variable with a significant contribution to the predictions for any of the five path-ways analyzed.

 RC:   In the Discussion section, lines 384-388, the authors state: “Using the developed open-source calculator, we could make individual predictions for each AF patient in whom EC is a therapy option and avoid the procedure in cases with a low likelihood of having a successful cardioversion and opt for a rate control strategy, which has been significantly associated with increase risk of adverse events[42], or to avoid antiarrhythmic drugs in patients who have high likelihood of spontaneous restoration of SR”. I find this very problematic, since this should be always in clinical practice refined. For example, a highly symptomatic patient presenting with atrial fibrillation in whom the sinus rhythm cannot be restored, should undergo a trial of antiarrhythmic therapy and then a new cardioversion attempt. In case of recurrence under antiarrhythmics catheter ablation of atrial fibrillation must be discussed. Furthermore, avoiding antiarrhythmic drugs in patients who have a high likelihood of spontaneous restoration of SR is not the best strategy, because of the nature of atrial fibrillation to become permanent. As such, strategies such as risk factor management, antiarrhythmic therapy and ablation need to be considered.

AA: We agree with the reviewer comments. The paragraph has been changed accordingly.

Round 2

Reviewer 1 Report

Currently, this article meets the criteria for publication in the JCM.

Best regards

Author Response

We thank the reviewer for his/her comment addressing that the manuscript currently meets the criteria for publication

Reviewer 2 Report

Dear Sir/Madam,

Thank you for reviewing the manuscript and addressing the mentioned issues.

However, further improvement can be made to the paper such as:

  1. The issue regarding the sample size: first of all, thank you for acknowledging the limitation. Secondly, you mention that “the number of events was enough to perform a proper evaluation of the machine learning models and is reflected in that [you] were able to ascertain statistically significant differences in performance between our developed models and the risk scores”. In order to make it easier for the reader to understand, please provide the learning curve of your machine learning model.
  2. Regarding the EHRA classification: please mention that providing the EHRA classification was not possible in your manuscript.

Best regards

Author Response

We would like to thank the Reviewer for the comment. In order to make it easier for the reader to understand the issue raised by the reviewer, we have added a figure (Figure 4) providing the learning curves for the machine learning models. (Page 8, Lines 311-317).

Regarding the EHRA classification suggestion, we have mention why we are not providing it in the new version of the manuscript (Page 200, Lines 200-201).